# Contrast-Enhanced Cardiac Computed Tomography and the Presence of Intravascular Air: A Patient Safety Study

**DOI:** 10.3390/jcm14144842

**Published:** 2025-07-08

**Authors:** Karim Bahadurali Samji, G. Sanjaya Chandrarathne, Wasim Khan, Hefin Jones, Richard Owen, Dilini Vethanayagam

**Affiliations:** 1Department of Radiology and Diagnostic Imaging, University of Alberta, Edmonton, AB T6G 2R3, Canada; ksamji@ualberta.ca (K.B.S.); hefin1@ualberta.ca (H.J.); rowen@ualberta.ca (R.O.); 2Edmonton HHT Center, Edmonton, AB T6G 2B7, Canada; galkotuw@ualberta.ca (G.S.C.); wakhan@ualberta.ca (W.K.); 3Department of Medicine, University of Alberta, Edmonton, AB T6G 2G3, Canada

**Keywords:** procedural air emboli, contrast-enhanced computed tomography, patient safety, quality improvement, risk management, right-to-left shunts

## Abstract

**Background/Objectives:** Air embolism on contrast-enhanced computed tomography (CECT) scans may have significant consequences, particularly if a right-to-left shunt is present, as seen in hereditary hemorrhagic telangiectasia. We sought to evaluate the frequency of CECT-associated air emboli in a single tertiary care referral center. **Methods:** Consecutive non-enhanced and contrast-enhanced cardiac CT studies (NECCT and CECCT, respectively) were evaluated prospectively over a 6-month period. Following the University of Alberta’s Health Research Ethics Board approval (code: Pro00042313; date: 1 May 2014), two experts reviewed all studies independently to assess for the presence and location of air emboli. The control group consisted of only NECCTs. All patients, except for the control group in this study, had an IV cannula placed. When present, the number, volume, and location of air emboli were recorded. **Results:** In this study, 110 subjects underwent intravenous cannula placement and both NECCT and CECCT. Of these, 27 of the NECCT studies (24.5%) and 36 of the CECCT studies (32.7%) demonstrated intravascular air emboli. Of those with air emboli, the average volume of intravascular gas was 19.22 ± 25.35 µL in the NECCT studies, with most of the intravascular air (70.4%) seen in the right atrial appendage (RAA). The average volume of intravascular air was 14.81 ± 26.54 µL in the CECCT studies, with most of the intravascular air also located within the RAA (72.2%). The incidence of intravascular air was higher in the CECCT group (28.6% increase), with lower volumes of intravascular air. None of the subjects in the control group (n = 28), who underwent NECCT without intravenous cannulation, demonstrated air emboli. **Conclusions:** Air emboli were present in a significant proportion of subjects undergoing intravenous cannulation and subsequent CECT. The use of CECT should be carefully considered in high-risk populations.

## 1. Introduction

Computed tomography (CT) scans have evolved from basic uses to very advanced technology since the inception of the modality in 1971 [1]. CTs now play a vital role in the diagnosis, monitoring, and follow-up of numerous conditions, including several cardio-thoracic and abdomino-pelvic conditions, for over 50 years, and many clinicians rely on the accuracy of this technology for patient care [1,2]. It is well known that contrast enhancement can improve diagnostic accuracy and aid in surgical planning [3]. Intravascular air embolism from contrast-enhanced CT (CECT) scans has been previously reported [4,5,6,7]. Intravascular air embolism is presumed to be a rare occurrence without significant clinical consequence; however, it is likely that they occur more frequently than reported [8,9]. Individuals with right-to-left shunts are particularly vulnerable to the deleterious consequences of iatrogenic air emboli.

Vascular air embolism occurs in CECT when air is introduced into the intravascular space through venous cannulation and/or intravenous contrast administration and can be transported to the cardiopulmonary vasculature if appropriate precautions are not adhered to, including patient positioning [4,5,6,7,8,9]. The resultant vascular physiologic effects depend on the volume and rate of air introduced. Small-to-moderate-sized air emboli have a reported incidence of 11–23% on CECT [7]. In most cases these air emboli are asymptomatic, but larger volumes can lead to hemodynamic sequelae and reduced cardiac output and can rarely be fatal. Air accumulation in the right ventricle may impede diastolic filling, and during systole, air is pumped from the right ventricle into the pulmonary arterial system. Air emboli may also reduce in size and may occlude progressively smaller vessels; this is particularly relevant in individuals with a right-to-left shunt (RLS), who are particularly vulnerable to blockages in the coronary and cerebral vasculature [10,11]. In general, 300–500 mL of air introduced at a rate of 100 mL/sec is considered a fatal dose for humans, precipitating cardiac arrest [12]. The severity of physiologic consequences depends on the volume of air, the rate of air entry, the type of gas, the patient’s position, the presence and severity of vascular shunts such as those seen with intrapulmonary and intracardiac shunts, and the patient’s general health status [13,14]. The effects will vary according to the vessels affected; however, they are predominantly cardiovascular, pulmonary, and neurologic [14].

Individuals with RLS pose a high risk of paradoxical embolism due to air entering the systemic circulation from the right heart; this is common in certain patient populations, including intracardiac RLS in those with unrepaired congenital heart lesions and intrapulmonary RLS due to pulmonary arteriovenous malformations (PAVMs), where direct communication between the pulmonary arterioles and venules results in loss of the fine pulmonary capillary beds where air emboli are normally filtered out [15]. As 80–90% of patients with PAVMs have hereditary hemorrhagic telangiectasia (HHT), clinician vigilance for iatrogenic air emboli is critical to ensure patient safety and avoidance of inadvertent neurovascular phenomena [16]. Air emboli are especially dangerous from a neurological perspective in HHT because, in addition to PAVMs, this disease predisposes patients to strokes and transient ischemic attacks due to cerebral arteriovenous malformations [17,18,19,20].

Given that certain patient populations with RLS [21] are at much greater risk for complications from iatrogenic air emboli, extra precautions are required when imaging modalities with the use of intravenous contrast enhancement are utilized [8,9,10]. Moreover, larger intravenous air emboli are often detected following the use of an automated power injector for the administration of contrast, which can result in increased morbidity and mortality; symptoms when they do occur are often nonspecific but can include nausea and vomiting, chest pain, acute onset of dyspnea, vertigo, loss of consciousness, seizure, focal paralysis, and loss of sensation in an extremity [22]. As air embolism can be potentially fatal, along with radiologists and radiology nurses, front-line clinicians, including physicians and nurse practitioners ordering these tests, should be well versed in preventing, identifying, and managing this complication [23]. If venous air embolism is suspected following contrast injection, along with the use of high-flow oxygen administration, Durant’s maneuver should be performed by placing the patient in the left lateral decubitus and Trendelenburg position [23,24,25]. One Austrian study by Groell and colleagues analyzed 677 patients with CECTs; air emboli were found in 79 patients (11.7%), with 7 exhibiting air emboli in multiple locations [8].

It is postulated that air emboli may enter the systemic circulation through the process of venous cannulation, connection of the contrast tubing to the intravenous cannula, and micro-bubbles in the contrast media. In another Austrian study, intravascular air was detected in 10 out of 208 (4.8%) patients who underwent CTs following venous cannulation, although attempts at measuring the volume of air emboli were unsuccessful [12].

The purpose of this study was to evaluate the prevalence of air emboli before and after contrast administration in a single tertiary/quaternary care center, a major site for cardiac surgery (Mazankowski Alberta Heart Institute) and the only HHT Center of Excellence in Western Canada, servicing many rural and remote communities with local CT imaging facilities to deliver care closer to home.

## 2. Methods

We compared the total volume and the anatomical location of air emboli introduced iatrogenically during non-enhanced and contrast-enhanced cardiac CT (NECCT and CECCT, respectively). Non-enhanced CT scans (in patients without intravenous cannulation) were used as controls.

A prospective single-center cohort study was conducted following institutional ethics board approval. A total of 110 consecutive cardiac CT (CCT) scans were evaluated prior to contrast administration (NECCT) and following contrast administration (CECCT). Intravenous cannulation was performed prior to NECCT, and the CECCT was performed immediately following NECCT. Additionally, a control group of 28 consecutive CCTs without contrast administration or intravenous cannulation was compared to the study group. De-identified scan data were utilized and stored on a secure, password-protected image database within the premises of the Department of Radiology and Diagnostic Imaging research office at the University of Alberta Hospital, Alberta Health Services.

Air embolism was defined as any focus of attenuation with an attenuation value of less than 900 Hounsfield Units (HU) within the systemic veins or the pulmonary arteries. The measurements were performed in the axial plane. The diameter of the air embolism was measured and subsequently converted to a volume, assuming the embolism was a sphere. The volume of each detected air embolism was then summed to give the total volume of air embolism.

### 2.1. Inclusion Criteria

All subjects aged 18 to 65 years and undergoing CCTs were captured over a consecutive 6-month period.

### 2.2. Exclusion Criteria

Patients with coronary stents, precordial surgical metallic clips, sternal wires, or any other metal implants that would interfere with interpreting the presence of intravascular air on the CCT scans were excluded. Pediatric (age less than 18) and pregnant subjects were also excluded.

Two expert reviewers assessed all CCT studies, both pre- and post-contrast enhancement (NECCT and CECCT, respectively), independently (blinded). They identified and measured the volume of intravascular air when present in each study. Discrepancies were adjudicated by a third, senior imaging expert. As a result of this process, 72 of 182 CCT studies were eliminated from the final results due to poor inter-rater reliability between the three reviewers. NECCTs from patients without venous cannulation were used as the control group.

### 2.3. Statistical Analysis

Descriptive statistical analysis was used to calculate the average air volume with standard deviation per subject and the number of intravascular air events (see Table 1).

## 3. Results

A total of 110 subjects underwent both NECCT and subsequent CECCT. 27 (24.5%) had intravascular air, with a subset of 8 (7.3%) showing air emboli in multiple locations on the NECCT portion of the examination, which was performed following intravenous cannulation. On the CECCT portion of the examination, 36 (32.7%) had evidence of intravascular air, with 9 (8.2%) showing air emboli in multiple locations (see Figure 1 and Figure 2). As demonstrated in Table 1, the average intravascular air volume among patients with emboli was 19.2 ± 25.4 µL and 14.8 ± 26.5 µL for the NECCT and CECCT studies, respectively.

Table 2 demonstrates the locations of intravascular air in the NECCT group. Air emboli were detected in the right atrial appendage (RAA), right internal mammary vein (RIMV), right ventricle (RV), right ventricle outflow tract (RVOT), and superior vena cava (SVC). The most common location for air volumes was the RAA/RA, with 21 (77.8%) out of 27 subjects demonstrating air emboli in RAA/RA. The highest average volume of air recorded per subject was found within the RIMV, at 35 ± 43.84 µL.

Table 3 demonstrates the locations of intravascular air in the CECCT group. Air volumes were found to be in similar locations to the NECCT group, but with a greater number of air emboli (45 subjects, in comparison to 35 subjects in the NECCT group). The greatest incidence of intravascular air was also found to be within the RAA, involving 27 patients (77.14%) and a total volume of 368 µL of intravascular air. The highest average volume of air attributed to a single patient was 35 ± 43.84 µL, found in the RIMV.

A total of 28 subjects underwent NECCT where venous cannulation was not performed and were taken to represent the control group. As demonstrated in Table 1, none of these patients had air emboli.

The incidence of intravascular air emboli increased post-contrast administration by about 2–3% on average, while 97–98% of the air introduced into patients occurred during placement of the venous cannula, before the introduction of any contrast agent.

## 4. Discussion

Air embolism is predominantly an iatrogenic complication that occurs when atmospheric air is introduced into the systemic circulation due to a relative negative pressure gradient within the vessel. Air emboli are usually considered asymptomatic and often go unreported. However, larger air emboli can contribute to increased risk and mortality. Our study demonstrates that air emboli are frequently present in patients undergoing CT, with the largest volume of air being introduced during venous cannulation. Air may enter the systemic circulation through cannulation of peripheral vessels, subsequent connection of tubing from the IV site to the pump injector containing the contrast media, and from pre-existing microbubbles in the contrast media itself.

Previous studies focused on the relationship of air emboli to the rate of flow and type of contrast media [9]. Our study focused on the differences in air emboli detected between subjects without intravenous cannulation (or contrast administration), those with intravenous cannulation but without contrast administration, and those with intravenous cannulation and contrast administration. As expected, our study found that patients without intravenous cannulation (or contrast administration) had no air emboli. In the 110 subjects with intravenous cannulation, 27 were found to have air emboli on the NECCT portion of this study, and 36 were found to have air emboli on the CECCT portion of this study.

The process of venous cannulation resulted in the presence of air emboli, independent of contrast administration, in most cases. The administration of contrast resulted in the introduction of further air emboli. The average volume of air per subject on CECCT was lower than on NECCT, measuring 14.8 µL and 19.2 µL, respectively, presumably secondary to dissemination of gas following contrast administration. The incidence of air emboli further increased from 35% to 45% following contrast administration. Most of the air emboli were located in the RAA/RA on both NECCT and CECCT. Previous studies have also shown a significant rate of air embolism following peripheral contrast injection for CT; for example, a study by Groell and colleagues demonstrated an incidence rate of 11.7% [8].

The asymptomatic and underreported nature of air emboli during venous cannulation and contrast administration has led to insufficiently standardized management of patients who may be at elevated risk for developing serious consequences from air emboli. These high-risk patient populations, such as those with HHT and pulmonary arteriovenous malformations, need careful monitoring. Consequences of air embolism may be associated with significant morbidity and mortality, including hypoxemia, cardiovascular collapse, and death. It is reported that symptomatic air emboli may give rise to a mortality rate of 30% if left untreated [7]. Patients with air emboli leading to hemodynamic compromise should be treated with hyperbaric therapy and/or 100% oxygen administration.

Study limitations include the relatively small cohort of subjects utilized and homogeneity from a single tertiary/quaternary care center; however, the latter allows for improved control of methodology, and therefore dissemination to other sites can benefit from this early information. This study is also limited by the lack of clinical follow-up of those patients with air emboli; however, as stated previously, air embolism following intravenous cannulation and contrast injection is usually asymptomatic. It can also be argued that as the sstudy population was not specifically high risk (i.e., those with RLS as seen with HHT), where air embolism could potentially be catastrophic, this does not necessarily affect the generalizability of our findings with regard to this high-risk population. Future studies, with a larger number of subjects, would help to confirm our findings and refine our knowledge of this phenomenon.

Although many cardiothoracic and other radiologists may be familiar with this phenomenon, other clinicians are often not aware of how imaging is requisitioned and how it may impact patient safety. Mitigation strategies to prevent air embolism, such as placing the patient supine during venous cannulation and flushing the lumen of the catheter tubing prior to connecting it with the patient’s intravenous cannula, should be further investigated to determine if they reduce the risk of air emboli. The incidence of air emboli with hand injections rather than with a power injector system, as well as technologist training specific to the procedure of cannulation, should also be further explored [26].

Our study demonstrated that air emboli are associated with both intravenous cannula placement and contrast administration. This is an important consideration when arranging diagnostic imaging in patients from certain high-risk groups, such as those with HHT, as a patient with a right-to-left shunt has an increased risk for developing symptomatic air emboli if precautionary measures are not followed. It is important to note that the placement of intravenous access lines is when most of the intravascular air is introduced, and therefore the decision to obtain intravenous access in high-risk patients should be considered very carefully. We recommend that patients with known right-to-left shunts have their initial intravenous access lines placed with air-eliminating filters (ideally 0.22-micron filters). Once the line is secured, the filter can be removed and connected to the automated power contrast injector. Though this second connection may introduce some intravascular air into the patient, this step only accounted for an additional 2–3% increase in intravascular air volume, on average, during our study, suggesting it is less likely to be significant.

In summary, from the perspective of procedural air emboli in non-high-risk individuals, contrast-enhanced CT studies are safe. However, in high-risk populations, particularly with RLS as seen with HHT, intravenous cannulation should be avoided if at all possible. If imaging tests are required in these individuals, an unenhanced examination should be performed wherever feasible. Particular care should be taken when IV cannula placement and contrast administration are planned for CT procedures to reduce the incidence of air emboli—especially in high-risk patient populations. Training in the CT suite for imaging technologists, nurses, and physicians may mitigate this issue—and may be more relevant to regional satellite hospital sites as we review significant needs to improve local access to care (including diagnostic imaging) for many individuals with HHT such that they can be served closer to home. This is particularly relevant in countries like Canada, the second largest country in the world in relation to geography, particularly in Western Canada, as the HHT Center serves a large proportion of rural and remote patients.

## Figures and Tables

**Figure 1 jcm-14-04842-f001:**
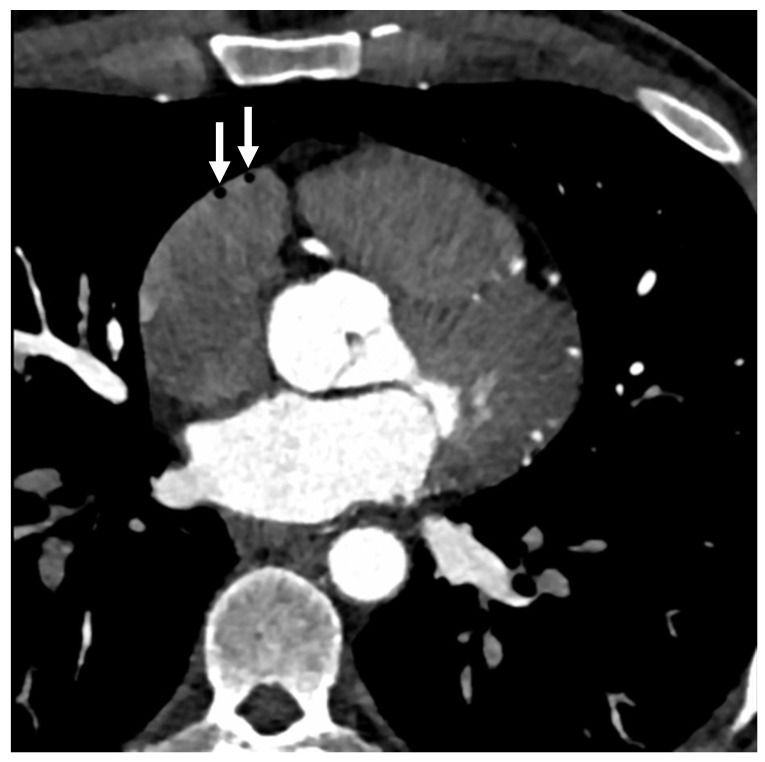
Cardiac computed tomography images with intravascular air in the right ventricle outflow tract. Note: White arrows show the intravascular air volumes in an individual who underwent contrast- enhanced chest CT.

**Figure 2 jcm-14-04842-f002:**
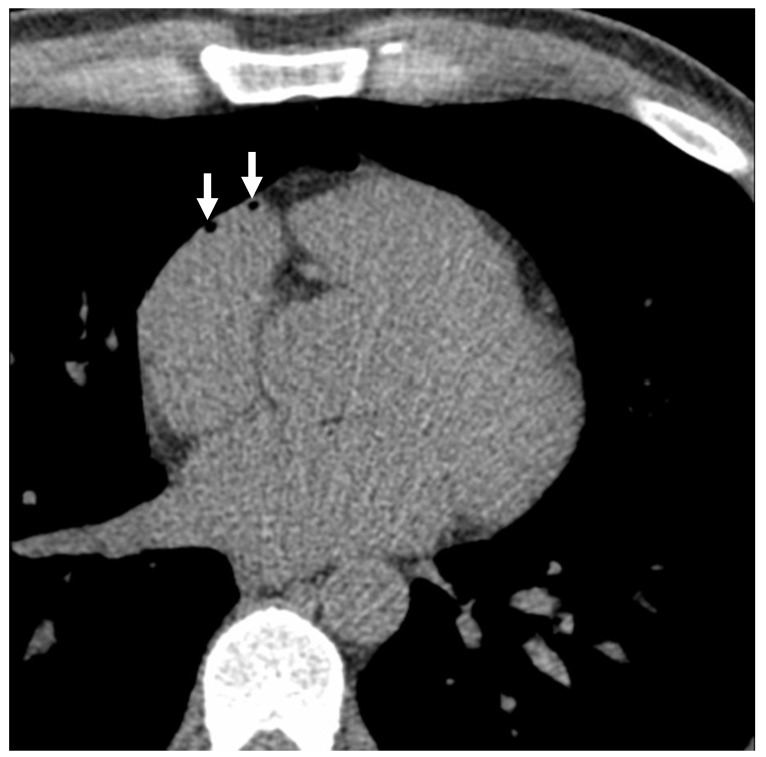
Cardiac computed tomography images with intravascular air in the right atrial appendage. Note: White arrows show the intravascular air volumes in a normal individual, prior to the administration of contrast, who underwent chest CT.

**Table 1 jcm-14-04842-t001:** Intravascular air volumes and type of CCT scans.

	NECCTs Performed Without Intravenous Cannulation	NECCTs Performed with Intravenous Cannulation	CECCTs
Number of subjects	28	110	110
Total number of subjects with air bubbles	0	27	36
Number of subjects with multiple locations of air bubbles detected	0	8	9
Total intravascular air volume (µL)	0	519	533
Average air volume per subject (µL)	0	19.22 ± 25.35	14.81 ± 26.54

IV—intravenous.

**Table 2 jcm-14-04842-t002:** Intravascular air volumes by location on NECCT.

Location	Number of Emboli by Location	Total Air Volume by Location (µL)	Average Air Volume by Location (µL)
RAA/RA	21	315	15.00 ± 25.89
RIMV	02	70	35.00 ± 43.84
RV	02	22	11.00 ± 9.89
RVOT	08	97	12.23 ± 10.48
RVOT/RV	01	14	14.00
SVC	01	01	01.00

RAA—right atrial appendages; RA—right atrium; RIMV—right internal mammary vein; RV—right ventricle; RVOT—right ventricle outflow tract; SVC—superior vena cava.

**Table 3 jcm-14-04842-t003:** Intravascular air volumes by location in CECCT.

Location	Number of Emboli by Location	Total Air Volume by Location (µL)	Average Air Volume by Location (µL)
RAA/RA	27	368	13.63 ± 27.95
RIMV	02	70	35.00 ± 43.84
RV	04	17	04.25 ± 4.03
RVOT	10	69	06.90 ± 5.53
RVOT/RV	01	05	05.00
SVC	01	04	04.00

RAA/RA—right atrial appendages/right atrium; RIMV—right internal mammary vein; RV—right ventricle; RVOT—right ventricle outflow tract; SVC—superior vena cava.

## Data Availability

The original contributions presented in this study are included in the article. Further inquiries can be directed to the corresponding author.

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
