# Peer review of "Contrast-Enhanced Cardiac Computed Tomography and the Presence of Intravascular Air: A Patient Safety Study"

_jcm, 2025, doi:10.3390/jcm14144842_

Round 1
Reviewer 1 Report
Comments and Suggestions for Authors
Please read the attachment. Thank you.

Please read the attachment. Thank you.
Author Response
Responses to comments from Reviewer 1
- NECCT is referred to as both a baseline and a comparator—ensure it's not
confusing.
We have tried to highlight this through multiple edits throughout the manuscript.
- Please define the parameters for "air embolism" (minimum volume, resolution
criteria).
We have included this in paragraph 3 of the “Methods” section.
- Please describe how air volume was measured (e.g., software, manual
estimation).
We have included this in paragraph 3 of the “Methods” section.
- Were both NECCT and CECCT performed on the same individuals? If so, the
time lag between the two should be mentioned.
Yes, these studies were performed on the same individuals, with CECCT performed immediately following NECCT. The intravenous cannula was placed prior to NECCT.
There was an additional set of control patients (n=28) that did undergo non-contrast CT without intravenous cannulation.
This information has been further clarified through edits in paragraphs 1 and 2 of the “Methods” section.
- Please state whether any clinical symptoms were monitored post-scan (even if
none were observed) and consider reorganizing for clarity: First present
demographics (age, sex, comorbidities if available), then incidence, volume, and
location.
We did not monitor for clinical symptoms post-scan. The study was primarily designed as an imaging evaluation.
⎯ Figure 1a should be Figure 1, and Figure 1b should be Figure 2.
This has been corrected.
- Please include statistical analysis: Was the difference between 24.5% vs. 32.7%
significant? What test was used?
We performed basic statistics on Excel.
- Please discuss limitations: Small sample size, single-center design, lack of
clinical follow-up.
This has been included/elaborated in paragraph 5 of the “Discussion” section.
- Please place findings in context: How does your data compare to previously
reported frequencies?
This comparison has been provided at the end of paragraph 3 of the “Discussion” section.
- Minor grammatical and stylistic improvements needed. For example,
✓ "None of the subject within the control group..." → "None of the subjects in the
control group..."
✓ Be consistent with tense and phrasing.
✓ "CECT should therefore only be undertaken..." – consider revising to a less
absolute recommendation, e.g., "should be carefully considered."
Thank you. These changes have been made as suggested.
Reviewer 2 Report
Comments and Suggestions for Authors
The manuscript is interesting. I have the comments and questions to the authors:
- About the first sentence in the introduction (line 36-37): if there is a reference source, it would be better to cite it.
- Line 54-55. Specify what is meant by the arterial system? During right ventricle systole, an air emboli is pumped into the pulmonary artery.
- Describes please the characteristics of the I/V cannulas used.
- Briefly describe please the protocol of CTA examine.
- The white arrows are not visible in the Figure 1. Add please arrows so that the reader can easily understand the locations of the embolies.
- Line 158. Inferior or internal mammary vein?
- Line 193. Provide please the references of previous studies.
- Line 214-215. “it is reported that symptomatic air emboli may give rise to a mortality rate of 30%” - provide please the reference.
- The authors’ point view is interesting: can the statistical data on I/V cannulation associated embolism in this study be generalized to I/V cannulation in general? If so, it would be better to briefly mention this in the discussion.
Author Response
Responses to comments from Reviewer 2
- About the first sentence in the introduction (line 36-37): if there is a reference source, it would be
better to cite it.
A reference has been provided.
- Line 54-55. Specify what is meant by the arterial system? During right ventricle systole, an air
emboli is pumped into the pulmonary artery.
This has been clarified in the revised manuscript.
- Describes please the characteristics of the I/V cannulas used.
Peripheral intravenous cannulas were utilized.
- Briefly describe please the protocol of CTA examine.
We used a routine institutional CTA protocol.
- The white arrows are not visible in the Figure 1. Add please arrows so that the reader can easily
understand the locations of the embolies.
The arrows have been added as requested.
- Line 158. Inferior or internal mammary vein?
This has been clarified in the revised manuscript.
- Line 193. Provide please the references of previous studies.
A reference has been provided.
- Line 214-215. “it is reported that symptomatic air emboli may give rise to a mortality rate of 30%” -
provide please the reference.
A reference has been provided.
- The authors’ point view is interesting: can the statistical data on I/V cannulation associated embolism
in this study be generalized to I/V cannulation in general? If so, it would be better to briefly mention
this in the discussion.
This has been included in the last paragraph of the “Discussion” section on the revised manuscript.
Round 2
Reviewer 1 Report
Comments and Suggestions for Authors
All concerns have been addressed and solved.
Thank you
Reviewer 2 Report
Comments and Suggestions for Authors
Dear authors, I have no more questions.